# Proteomic Insights into Seminal Plasma and Spermatozoa Proteins of Small-Spotted Catsharks, *Scyliorhinus canicula*: Implications for Reproductive Conservation in Aquariums

**DOI:** 10.3390/ani14091281

**Published:** 2024-04-24

**Authors:** Marta Muñoz-Baquero, Laura Lorenzo-Rebenaque, Ximo García-Domínguez, Jesús Valdés-Hernández, Daniel García-Párraga, Clara Marin, Francisco Alberto García-Vázquez, Francisco Marco-Jiménez

**Affiliations:** 1Department of Animal Production and Health, Veterinary Public Health and Food Science and Technology, Biomedical Research Institute, Faculty of Veterinary Medicine, Cardenal Herrera-CEU University, CEU Universities, Calle Santiago Ramón y Cajal 20, 45115 Alfara del Patriarca, Spain; marta.munozbaquero@uchceu.es (M.M.-B.); clara.marin@uchceu.es (C.M.); 2Fundación Oceanogràfic de la Comunidad Valenciana, 46005 Valencia, Spain; dgarcia@oceanografic.org; 3Institute for Animal Science and Technology, Universitat Politècnica de València, 46022 Valencia, Spain; laulore@upv.es (L.L.-R.); ximo.garciadominguez@gmail.com (X.G.-D.); jesusvaldeshernandez@yahoo.com (J.V.-H.); 4Veterinary Services, Avanqua-Oceanogràfic S.L., Ciudad de las Artes y las Ciencias, 46013 Valencia, Spain; 5Departamento de Fisiología, Facultad de Veterinaria, Universidad de Murcia, Campus de Excelencia Internacional Mare Nostrum, 30100 Murcia, Spain; fagarcia@um.es

**Keywords:** Scyliorhinidae, Elasmobranchii, spotted dogfish, sperm, conservation biology

## Abstract

**Simple Summary:**

This study introduces the first proteome database for shark semen, focusing on small-spotted catsharks. It explores seminal plasma and spermatozoa proteomic profiles, uncovering protein differences between wild and aquarium populations. Key findings include 305 seminal plasma and 535 spermatozoa proteins, with significant variations in spermatozoa proteins that could influence reproductive success. This work lays the groundwork for identifying fertility biomarkers in shark conservation efforts.

**Abstract:**

In the ex situ conservation of chondrichthyan species, successful reproduction in aquaria is essential. However, these species often exhibit reduced reproductive success under human care. A key aspect is that conventional sperm analyses do not provide insights into the functional competence of sperm. However, proteomics analysis enables a better understanding of male physiology, gaining relevance as a powerful tool for discovering protein biomarkers related to fertility. The present work aims to build the first proteome database for shark semen and to investigate the proteomic profiles of seminal plasma and spermatozoa from small-spotted catsharks (*Scyliorhinus canicula*) related to the underlying adaptations to both natural and aquarium environments, thereby identifying the reproductive impact in aquarium specimens. A total of 305 seminal plasma and 535 spermatozoa proteins were identified. Among these, 89 proteins (29.2% of the seminal plasma set) were common to both spermatozoa and seminal plasma. In the seminal plasma, only adenosylhomocysteinase protein showed differential abundance (DAP) between wild and aquarium animals. With respect to the spermatozoa proteins, a total of 107 DAPs were found between groups. Gene Ontology enrichment analysis highlighted the primary functional roles of these DAPs involved in oxidoreductase activity. Additionally, KEGG analysis indicated that these DAPs were primarily associated with metabolic pathways and carbon metabolism. In conclusion, we have successfully generated an initial proteome database for *S. canicula* seminal plasma and spermatozoa. Furthermore, we have identified protein variations, predominantly within spermatozoa, between aquarium and wild populations of *S. canicula*. These findings provide a foundation for future biomarker discovery in shark reproduction studies. However, additional research is required to determine whether these protein variations correlate with reproductive declines in captive sharks.

## 1. Introduction

The development of reproductive technologies is essential for the conservation of chondrichthyan species considered threatened or endangered [1,2,3], particularly those facing challenges in reproducing under aquarium conditions [4]. Comparative studies on the reproductive physiology of wild and aquarium animals could potentially advance the application of reproductive technologies [4,5]. This knowledge is crucial for understanding their reproductive aspects and establishing suitable ex situ conditions in aquaria to complement in situ conservation programs in the wild [6]. However, a limited number of studies have evaluated their reproductive differences when modifying in habitats in aquaria [4,7]. Consequently, although the use of artificial insemination is gaining increasing attention in chondrichthyans [4,5], the production of offspring has been achieved anecdotally [5,8,9].

Conventional sperm analyses, such as concentration, motility, morphology, and viability [4,7,10,11,12,13], do not provide enough information about the functional competence of spermatozoa [14,15] and do not clearly explain the subcellular factors associated with limited reproductive success [16]. Proteomics has emerged as a promising tool for addressing fundamental questions regarding the composition and function of sperm cells, and for identifying altered proteins and pathways in infertile males’ seminal fluid and sperm cells [16,17,18,19]. Proteomic approaches have been applied in the sperm of aquatic animals, teleost fish, and marine mammals [17,20,21,22,23,24,25]. However, when it comes to chondrichthyans, limited data are available for sperm proteins [26,27,28]. To the best of our knowledge, no previous studies have analyzed the composition of seminal plasma proteins in chondrichthyans. In fact, *Scyliorhinus canicula, Linnaeus 1978,* is an interesting model to explore differential changes in the abundance of such proteins, due to its easy handling and accessibility in aquaria and fish local markets, small size and adequate conservation status.

In this study, we aimed to identify and characterize seminal plasma and spermatozoa proteins in *S. canicula*, thereby enhancing our understanding of omics data concerning shark semen and chondrichthyan species in general. Additionally, we assessed the utility of protein composition analysis as a diagnostic tool for detecting potential reproductive modifications in *S. canicula* under human care.

## 2. Materials and Methods

### 2.1. Ethics

Wild individuals involved in the experiment were obtained from local fisheries as a result of accidental captures. The experimental portion of the current study involving aquarium animals was approved by the Ethical Animal Experiments Committee at Oceanogràfic Valencia (Reference number: OCE-18-19) for the protection of animals used for scientific purposes. The study complied with the regulations and policies of Oceanogràfic-Valencia and adhered to the Canadian Council on Animal Care documents.

### 2.2. Semen Origin

Eight seminal plasma and spermatozoa samples stored at −80 °C were thawed for the purpose of this study. These samples were obtained from a prior experiment that involved the comparative study of semen parameters in *S. canicula* [7]. Briefly, a total of 25 *S. canicula* males were used, consisting of 18 wild individuals donated from local fisheries from the Region of Valencian Community (Spain) and 7 aquarium individuals at Oceanogràfic of Valencia (Valencia, Spain). All animals from the study were classified as adult, displaying calcified claspers ensuring reproductive maturity [29]. Sample collection was performed from November 2019 to March 2020. Aquarium individuals were handled and maintained in dorsal recumbency (tonic immobility), inducing a physiological state of slight sedation. The cloaca was maintained out of the water, and the area was rinsed with sterile Shark’s Ringer solution (22 g/L urea and 9 g/L NaCl, [30]), and the surface was cleaned with sterile gauzes to prevent contamination of the external part of the ampullae pore. Semen samples were collected by applying direct pressure on the ampulla of the vas deferens by stripping. Semen was collected from the papilla of the ampulla using a 5 mL syringe. The first portion of the sperm sample was discarded, as were the samples polluted with urine or feces. Semen samples were transported directly to the laboratory within one hour of collection and maintained in a dark container at 4 °C until assessment. Wild individuals were provided by local fishermen and transferred to the laboratory within 4 h post-fishing. Semen samples from these individuals were collected from the ampulla following the same procedure and criteria as previously described. Similar laboratory conditions were applied to both groups during processing. A total of 8 pooled samples were established, comprising 4 from wild animals and 4 from aquarium animals, each created by mixing 100 µL from individual samples. The 4 pooled samples from the wild animals encompassed 5, 3, 3, and 6 animals from the ports of Valencia, Cullera (2 pooled samples), and Jávea, respectively. Regarding the aquarium animals, due to a limited number of specimens (n = 7), 4 pooled samples were generated. Two of these pooled samples each included 3 animals. The remaining pooled samples comprised the remaining animals, one from the initial sampling session and one from the second group exhibiting excess semen production (over 200 µL). An aliquot from each pool was taken to determine sperm quality as described in the following section. The remaining pooled samples were centrifuged at 7400× *g* for 10 min at 4 °C (Centrifuge 5810 R, Eppendorf, Fisher Scientific, Madrid, Spain). The supernatant was separated from the cell part, splitting the sample into two fractions: seminal plasma and spermatozoa. The seminal plasma fraction was verified under a microscope to ensure no spermatozoa were present and was directly flash-frozen (cryotubes) in liquid nitrogen and stored at −80 °C until use for the proteomic study. The pellet containing spermatozoa was immediately washed with shark’s Ringer solution (7400× *g* for 10 min at 4 °C, Centrifuge 5810 R, Eppendorf) to remove any remaining seminal plasma. Once centrifuged, the supernatant was discarded, and the pellet (cell fraction) was directly flash-frozen (cryotubes) in liquid nitrogen and stored at −80 °C until use.

### 2.3. Sperm Quality Assessment

Sperm concentration was determined using a Makler counting chamber after diluting the semen with Shark’s Ringer solution (1:100). Motility was assessed using 5 μL of diluted semen (1:20 in Shark’s Ringer solution). Total motility (%) was defined as the percentage of motile sperm, including spermatozoa vibrating without moving forward (200× magnification, phase contrast Nikon E 400, Izasa, Madrid, Spain) (Wyffels et al., 2020) [4]. Sperm viability (%) was assessed by incubating diluted sperm suspensions (1:20) in the dark for 10 min with 1 μL SYBR-14 and 1 μL propidium iodide (LIVE/DEAD Sperm Viability Kit L-7011, Thermo Fisher Scientific, Madrid, Spain) and counting at least 100 cells using an epifluorescence microscope (Nikon E 400). Spermatozoa with green fluorescence over the head region were assessed as having plasma membranes intact, and sperm with partially or totally red heads were assessed as having plasma membrane damage (Figure 1). Sperm mitochondrial membrane potential was assessed by incubating Shark’s Ringer solution-diluted sperm suspensions (1:20) in the dark for 30 min with 2 μL of carbocyanine dye (JC-1 Dye, Thermo Fisher Scientific, Madrid, Spain) mixed with 100 μL diluted sperm, and counting at least 100 cells using an epifluorescence microscope (Nikon E 400).

### 2.4. Comparative Proteomic Analysis: Sampling, Protein Extraction, and Quantification

Proteomic analyses were performed at the Proteomics Unit of the University of Valencia (SCSIE), Valencia, Spain (PRB2-ISCIII ProteoRed Proteomics Platform). Proteins from spermatozoa were extracted using Ringer’s lysis solution (22 g/L urea and 9 g/L NaCl). After mixing with lysis buffer, samples were sonicated for 5 min and centrifuged for 5 min at 15,870× *g* at 4 °C and diluted 1 to 10 with 50 mM ABC. The protein concentration from every sample was determined by a Protein Quantification Assay Kit (Macherey-Nagel, Düren, Germany).

### 2.5. Complete Proteome: Spectral Library Building by In-Gel Digestion and LC-MS/MS—Data-Dependent Acquisition Analysis

A data-dependent acquisition analysis was performed by building up a spectral library using in-gel digestion and LC-MS/MS and building the spectral library from a 1D SDS PAGE gel. A SWATH analysis of individual samples was performed to determine the quantitative differences between spermatozoa and seminal plasma protein composition among aquarium and wild *S. canicula* experimental groups, following procedures previously described by Garcia-Dominguez et al. [31]. 

The protein concentration obtained from both, spermatozoa and seminal plasma samples was determined by ProteinQuantification Assay Kit (Macherey-Nagel, Düren, Germany), according to the manufacturer’s instructions. 

For Spectral Library Building, a pool of 30 μg from equivalent aliquots of the samples, of the same type, were mixed, and sample loading buffer (appropriate volume of 4× Laemmli Sample Buffer with mercaptoethanol was added. The denaturation of the pools was carried out at 95 °C for 5 min and loaded into 1D PAGE. A 12% precast gel (Bio-Rad, Hercules, CA, USA) at 200 V for 5 min was used to perform the electrophoresis. The gel was fixed with 40% ethanol/10% acetic acid for an hour and stained with colloidal Coomassie (Bio-Rad, Hercules, CA, USA) for an additional hour. The gel was then washed and distained with H_2_O milliQ. Each gel lane corresponding to an experimental condition (aquarium vs. wild sharks and seminal plasma vs. spermatozoa) was divided into five equivalent slices. The gel slices were digested using 600 ng of trypsin (Promega, Madison, WI, USA) at 37 °C to build the spectral library. The trypsin digestion was stopped with 10% trifluoroacetic acid (TFA), and the supernatant was removed. Then, the library gel slices were dehydrated with pure acetonitrile (ACN). The resulting peptide solutions were combined with the corresponding supernatant and dried in a rotatory evaporator. The dried peptide mixtures were resuspended with 10–20 μL 2% ACN; 0.1% TFA. 

In parallel, for differential expression analysis 20 μg of individual samples from spermatozoa and seminal plasma of each group were set to 20 μL of 50 mM ammonium bicarbonate (ABC) and processed in solution. After cys reduction at 37 °C and alkylation, proteins were digested with 800 ng of trypsin. The sample digestion was stopped with 10% trifluoroacetic acid (TFA) to a final concentration of 1%. Final tryptic peptide solutions were at 0.31 μg/μL. 

#### 2.5.1. Spectral Library Acquisition

The analysis was performed using 5 μL of peptide fractions from in-gel digestion and loading it into a trap column (3 μm particle size C18-CL, 350 μm diameter × 0.5 mm long; Eksigent Technologies, Dublin, CA, USA). The samples were desalted with 0.1% TFA at 5 μL/min for 5 min for liquid chromatography and tandem mass spectrophotometry (LC-MS/MS). The peptides were then loaded into an analytical column (LC Column, 3 μm C18-CL, 0.075 × 150 mm, Eksigent Technologies, Dublin, CA, USA) and equilibrated in 5% ACN, 0.1% formic acid (FA). Peptide elution was carried out with a linear gradient of 7 to 40% B for 20 min (A: 0.1% FA; B: ACN, 0.1% FA) at a flow rate of 300 nL/min. The peptides were analyzed in a mass spectrometer nanoESI qQTOF (6600plus TripleTOF, ABSCIEX, Alcobendas, Madrid, Spain). Samples were ionized in a Source Type: Optiflow< 1 µL Nanoapplying 3.0 kV to the spray emitter at 200 °C. The tripleTOF was operated and the analysis was carried out in a data-dependent mode MS1, scanning from 350 to 1400 *m*/*z* for 250 ms. The quadrupole resolution was performed at ‘LOW’ for MS2, followed by 25 ms product ion scans from 100 to 1500 *m*/*z* in “high sensitivity”. Collision energy was set for all ions to optimum charge 2+ to 4+ ions. Up to 100 ions were selected for fragmentation after each survey scan. Dynamic exclusion was set to 15 s. The system sensitivity was controlled by analyzing 0.5 μg of K562 trypsin digestion (Sciex, Framingham, MA, USA). 

#### 2.5.2. SWATH-MS

For differential expression analysis 4 µL of individual digested samples were analyzed in the same LC-MS system as for Spectral Library Analysis with SWATH (DIA) acquisition. However, the elution gradient lasted for 45 min. The tripleTOF was operated in swath mode, in which a 0.050 s TOF MS scan from 350 to 1250 *m*/*z* was performed, followed by 0.020 s product ion scans from 350 to 1250 *m*/*z* on the 100 variable width windows from 400 to 1250 Da (3.05 s/cycle). 

### 2.6. Protein Identification, Validation and Quantification

For Spectral Library building, all DDA data from pooled samples were processed in combination using ProteinPilot software v5.0 (AB SCIEX, Alcobendas, Madrid, Spain) to identify a direct ortholog for each protein. The Paragon algorithm [32] of ProteinPilot v5.0 was used to search against the Uniprot Chordata (UniprotChordata_200721601.fasta) protein sequence database (562246 proteins searched) with the following parameters: trypsin specificity, cys-alkylation, no taxonomy restriction, and the search effort set to rapid with FDR (False Discovery Rate) analysis for proteins. A spectral library of 3010 proteins was obtained. The wiff files obtained for individual samples from the SWATH experiment were obtained by PeakView^®^ using the spectral library generated in the study and analyzed using MarkerView^®^ (v1.2, AB SCIEX, Alcobendas, Madrid, Spain). 

### 2.7. Proteome Statistical Analysis and Functional Annotation of the Differentially Abundant Proteins

The protein areas calculated were normalized by summing the total areas of all quantified proteins by MarkerView. The identification of the differentially abundant proteins (DAPs) among groups (wild vs. aquarium) was performed using a *t*-test. Proteins were considered differentially abundant if they exhibited a ≥2-fold change and *p*-value < 0.05. The whole Scyliorhinus torazame genome (scyTo version 1.0) was selected as a homologous reference organism (taxonomy ID 75743). Principal component analysis (PCA) and Heat-Map clustering were conducted using ClustVis (https://biit.cs.ut.ee/clustvis/, accessed on 1 March 2024). DAPs were submitted to the publicly available program STRING (http://string-db.org/, accessed on 1 March 2024) for a search of multiple proteins by names. To ascertain the KEGG pathway-enriched genes and the potential GO (Gene Ontology) classification, terms approximating biological process, molecular functions, and signaling pathways concerning KEGG pathways were used. Significant enrichment of Gene Ontology (GO) terms was identified based on the criteria of *p*-value < 0.05 and FDR-adjusted *p*-value < 0.05. In addition, default settings were used, with a confidence of 0.4 as the minimum required interaction score. The protein–protein interactions were predicted using the Search Tool for the retrieval of interacting proteins, which determines both physical and functional associations between proteins. 

### 2.8. Statistical Analysis 

Bayesian statistics were used to measure the relevance of the differences in sperm in vitro quality between the experimental groups (wild and aquarium groups). Hence, a model with a single effect of ‘treatment’ and flat priors was fitted. The marginal posterior distribution of the unknowns was performed with MCMC (Gibbs sampling) using four chains with a length of 50,000 iterations, a lag of 10, and a burn-in of 1000 iterations. The posterior mean of the differences in genera or metabolite abundances was estimated as the mean of the marginal posterior distribution of differences between the control and each of the treatments. These differences estimates were reported as units of standard deviations (SD) of each trait. The differences between experimental groups were considered relevant when these differences were higher than 0.5 units of SD, and the probability of the differences being higher (if the difference is positive) or lower (if negative) than 0 (P0) was higher than 0.9 [33].

## 3. Results

### 3.1. Sperm Quality Assessment

The descriptive statistics for sperm parameters of the two groups are presented in Appendix A. Significant differences were observed between the wild and aquarium groups for all sperm parameters except viability (Table 1). Aquarium animals consistently exhibited higher values for total sperm motility and mitochondrial membrane potential, whereas wild animals had a higher sperm concentration. The number of spermatozoa used for proteomic analysis ranged between 23.2–56.8 × 10^6^ for wild animals and between 24–37.6 × 10^6^ for aquarium animals.

### 3.2. Comprehensive Proteomic Analysis of Seminal Plasma and Spermatozoa

A total of 305 and 535 proteins were detected in all seminal plasma and spermatozoa samples, respectively. PCA was performed for both cohorts, showing differentiated sample clusters based on the seminal plasma and spermatozoa proteins from semen collected from aquarium and wild animals (Figure 2). Concerning the quantified seminal plasma proteins, the first two components of the PCA explained 100% of the total variance (PC1 68.6% and PC2 31.4%), discriminating between the four replicates of the aquarium and those of the wild (Figure 2A). The same analysis for spermatozoa showed 54.2% of the total variance (PC1 33.6% and PC2 20.6%) (Figure 2B). The PCA result for seminal plasma proteins did not show such clear clustering because scarce differences were observed between groups. Heat map analysis corroborated the PCA analysis, showing small differences clustered together for seminal plasma (Figure 2C), while they were clearly marked for spermatozoa proteins (Figure 2D). 

The completed detailed list of identified proteins for each experimental group is provided in Appendix A. To analyze differences between individuals in their natural environment and those in an aquarium environment, we compared the protein expression profiles of seminal plasma and spermatozoa from *S. canicula*. Out of the 305 proteins detected in the seminal plasma, 2 were identified as DAPs. Among the 535 proteins detected in the spermatozoa, 107 were identified as DAPs. In the aquarium group, there were 75 proteins with increased abundance and 32 proteins with decreased abundance compared with the wild group (Figure 3). 

Out of the proteins identified as DAPs, 2 in seminal plasma and 107 in spermatozoa, 1 and 42, respectively, were not present in the *Scyliorhinus torazame* UNIPROT database. Furthermore, 6 DAPs in the spermatozoa were uncharacterized. Consequently, a total of 59 DAPs present in the *Scyliorhinus torazame* UNIPROT database underwent functional annotation, as detailed in Table 2.

### 3.3. Seminal Plasma Differentially Abundant Proteins and Functional Annotation Analysis 

A single protein was identified with a protein ID probability exceeding 99%, along with two different peptides that had a minimum peptide ID probability of 95%, thus meeting the criteria for identification. Adenosylhomocysteinase (A0A401PRC8) was the only differentially abundant protein in aquarium animals, exhibiting an FC of 2.01 present in the *Scyliorhinus torazame* UNIPROT database (Appendix A). Since only a small number of differential proteins were found, no functional annotation analysis was performed.

### 3.4. Spermatozoa Differentially Abundant Proteins and Functional Annotation Analysis 

Among the 59 total DAPs identified in spermatozoa, 36 proteins exhibited increased abundance in the aquarium individuals, while 23 proteins exhibited increased abundance in the wild individuals (Table 2). By analyzing BP, we found that the DAPs from the complex PPI network were enriched in cellular processes (GO:0009987), small molecule metabolic processes (GO:0044281), carboxylic acid metabolic processes (GO:0019752), generation of precursor metabolites and energy (GO:0006091), energy derivation by oxidation of organic compounds (GO:0015980), aerobic respiration (GO:0009060), tricarboxylic acid cycle (GO:0006099), cellular aldehyde metabolic processes (GO:0006081), and glycerol catabolic processes (GO:0019563). The Gene Ontology MF analysis revealed the involvement of DAPs in catalytic activity (GO:0003824), oxidoreductase activity (GO:0016491), identical protein binding (GO:0042802), oxidoreductase activity, acting on the CH-OH group of donors, NAD or NADP as acceptor (GO:0051287), hydro-lyase activity (GO:0016836), and 3-hydroxyacyl-CoA dehydrogenase activity (GO:0003857). In addition, the KEGG pathway enrichment analysis revealed the association of the DAPs with metabolic pathways (map01100), carbon metabolism (map01200), fatty acid metabolism (map01212), and lysine degradation (map00310). The annotated results for the following terms were tabulated (Table 3 and Appendix A).

### 3.5. Establishment of PPI Networks and Module Analysis

We conducted a network analysis to investigate the interactions among the DAPs in spermatozoa using STRING (Figure 4). Our analysis revealed that out of the 59 proteins, 43 of them exhibited interactions with each other, forming a network with a total of 128 edges. Notably, the proteins with the highest degree of connectivity had a clustering coefficient of 0.452, and the *p*-value for the enrichment of protein-protein interactions was <1.0 × 10^−16^. These proteins were found to be associated with oxidoreductase activity, carbon metabolism, and metabolic pathways.

## 4. Discussion

In this study, we conducted a comprehensive analysis of the global proteome in seminal plasma and spermatozoa derived from both wild and aquarium *S. canicula*. Our research aims to deepen our insights into how aquarium conditions may influence the quality and fertilization potential of sperm, which holds significant importance for optimizing reproductive techniques. In aquarium facilities, the reduced quantity of external natural stimuli could potentially impact the cyclical reproduction patterns [4,34]. In our previous study, aquarium individuals showed higher total sperm motility, with no observed differences in sperm viability, mitochondrial membrane potential, and membrane integrity [7]. The potential reasons for variations in fish sperm quality between habitats may be attributed to factors such as water temperature, salinity, stocking density, sex ratio, or diet [35,36,37,38,39]. However, these conventional analyses do not provide insights into the functional competence of spermatozoa [40]. Assessing sperm quality and potential fertility poses challenges due to inherent factors related to the animals and extrinsic elements, such as environmental influences [41]. In this context, the identification of proteins responsible for cellular functions in cells and tissues has emerged as a crucial area of research for identifying fertility-associated biomarkers [16,19,42]. Comparative analyses of sperm proteomes have significantly contributed to our understanding of how spermatozoa acquire their fertilization capacity and why they exhibit varying levels of fertility [40,42,43].

The analysis of the shark proteome conducted in the present study allowed the identification of 305 and 535 proteins in seminal plasma and spermatozoa, respectively. To our knowledge, little importance relies on the 89 proteins that are simultaneously present in seminal plasma and spermatozoa. Similar phenomena have been observed in previous studies involving mammalian species [44,45,46]. This result currently represents the most extensive dataset of proteins in seminal plasma and spermatozoa from sharks, providing the first comprehensive characterization of the seminal plasma and spermatozoa proteomes in *S. canicula*, encompassing individuals inhabiting their natural habitat and those under aquarium conditions. However, this study also highlights the lack of information on these species. Among the 2 DAPs identified in seminal plasma, only 1 has an ortholog (in either one-to-one or one-to-many relationships) in *Scyliorhinus torazame*. Conversely, of the 107 DAPs found in the spermatozoa, only 59 have orthologs in *Scyliorhinus torazame*. Specifically, for the spermatozoa, 42 DAPs did not have any orthologs in other species.

In our current research, we identified only one protein as a DAP in the seminal plasma, the adenosylhomocysteinase (AHCY), which exhibited increased abundance in individuals from the aquarium. AHCY is a methyltransferase with catalytic activity, participating in amino acid biosynthesis [47,48]. It plays a pivotal role in the one-carbon metabolic cycle, a fundamental metabolic process that facilitates the transfer of one-carbon units essential for various biosynthetic processes (e.g., purines and thymidine), maintenance of amino acid levels (including cysteine, serine, and methionine), regulation of cellular redox balance, and epigenetic control [49]. Notably, AHCY is one of the most highly conserved enzymes found in a wide range of living organisms, spanning from bacteria, nematodes, and yeast to plants, insects, and vertebrates [50]. its crucial role in intermediary metabolism has been well-established [51], and it has also been identified in various fish species [23,52,53]. As AHCY plays a crucial role in intermediate metabolism [51] and is associated with spermatozoa motility, this result could be related to the increased spermatozoa motility in aquaria [23,51,52,53]. In line with this, our proteomics results indicate that aquarium conditions induce minimal modifications in seminal plasma composition compared to wild conditions. 

To explore the involvement of the 59 identified DAPs in BP, MF, and molecular pathways of wild and aquarium *S. canicula*, we used GO and KEGG enrichment to determine the functional annotation of these proteins. We found that these DAPs were primarily enriched in cellular processes, small molecule metabolic processes, carboxylic acid metabolic processes, generation of precursor metabolites and energy, energy derivation by oxidation of organic compounds, aerobic respiration, tricarboxylic acid cycle, cellular aldehyde metabolic processes and glycerol catabolic processes. The analysis of MF from GO showed that the DAPs were significantly enriched in catalytic activity, oxidoreductase activity, identical protein binding, oxidoreductase activity, acting on the CH-OH group of donors, NAD or NADP as acceptor, NAD binding, hydro-lyase activity, and 3-hydroxyacyl-CoA dehydrogenase activity. Similarly, the analysis of KEGG pathway enrichment showed that the DAPs are involved in metabolic pathways, carbon metabolism, fatty acid metabolism and degradation, valine, leucine, and isoleucine degradation, lysine degradation, butanoate metabolism, citrate cycle (TCA cycle), tryptophan metabolism, glyoxylate and dicarboxylate metabolism, pyruvate metabolism, propanoate metabolism, synthesis and degradation of ketone bodies, 2-oxocarboxylic acid metabolism, metabolism of xenobiotics by cytochrome P450, drug metabolism—cytochrome P450, biosynthesis of amino acids, fatty acid elongation, and terpenoid backbone biosynthesis. Interestingly, two matching proteins (A0A401NWX3 and A0A401P3W7) were included in three out of six significant functional categories, among which an enrichment of both genes was found in MFs (GO:0016491 and GO:0016616) and KEGG (map01100 and map01200) pathways. In addition, A0A401P5Y1 and A0A401QCT0 proteins overlapped between the MF and KEGG. On the one hand, it is noteworthy that MFs were predominantly related to oxidoreductase activity. On the other hand, classification analysis revealed that these pathways were mainly concentrated in metabolic pathways and carbon metabolism (KEGG). The DAPs among sperm proteins included key enzymes of the citric acid cycle (e.g., malate dehydrogenase, A0A401NWX3) and lipid synthase activity (fatty acid synthase complex, A0A401P5R6). These proteins have previously been detected in spermatozoa of teleost fish [54] and sea bream semen (*Sparus aurata*) [55]. Notably, malate dehydrogenase (MDH, A0A401NWX3), Superoxide dismutase (SOD, A0A401PG28), and thioredoxin domain-containing protein-2 (TXNDC2, A0A401PCP3) exhibited increased abundance in individuals from the aquarium. The MDH protein plays a crucial role in producing the energy required to support sperm motility. To do this, MDH facilitates the conversion between oxaloacetate and malate, a reaction pivotal to cellular metabolism. This process involves easily detectable cofactor oxidation/reduction, as highlighted by Musrati et al. [56]. In rainbow trout, a negative correlation was found between MDH activity and fertilization rate, suggesting that low-quality semen might have heightened energy needs [57]. Conversely, elevated MDH activity may negatively react to anoxic conditions, particularly during storage, or result from insufficient energy availability. Additionally, high-quality sturgeon sperm overexpress proteins like MDH, which is crucial for energy support via the citrate cycle [58]. This suggests that lower MDH expression in spermatozoa could impair sperm motility and, consequently, fertility. For instance, diminished MDH abundance in cryopreserved sea bream sperm was linked to decreased motility post freeze-thaw procedure [59]. Moreover, the presence of certain DAPs related to oxidative stress may further indicate potential membrane compromise. SOD1 and TXNDC2 potentially play an important role in regulating the redox status [60,61]. SOD1 is tightly associated with sperm quality, including sperm motility [62,63,64]. For example, in Sea Bream, the level of sperm SOD protein significantly increased during the cryopreservation process as a consequence of oxidative stress [59]. Thus, the observed increase in SOD1 and TXNDC2 levels in the sperm of aquarium individuals could indicate higher oxidative stress in these animals compared to wild individuals. 

The results of this study show that the predominant DAPs identified were metabolic enzymes, such as glycerol-3-phosphate dehydrogenase (GPD2, A0A401NP72), isocitrate dehydrogenase 3 (IDH3), fatty acid synthase (FASN, A0A401P5R6), and clusterin (CLU, A0A401PF86). GPD2, an enzyme responsible for catalyzing the conversion of glyceraldehyde 3-phosphate to 1,3-bisphosphoglycerate, has been observed to be overexpressed in high-quality sturgeon spermatozoa [58]. Additionally, this protein is recognized for facilitating tyrosine phosphorylation during the sperm capacitation process [65]. Furthermore, the critical role of glycolysis in sperm and its dependence on this sperm-specific enzyme suggest that GPD2 is a potential target for contraception. Hence, mutations or environmental agents that disrupt its activity could lead to male infertility [66]. Additionally, IDH3, a key enzyme in the mitochondrial tricarboxylic acid (TCA, map00020) cycle that catalyzes the decarboxylation of isocitrate to α-ketoglutarate while converting NAD+ to NADH, was down-regulated in the aquarium individuals. In humans, a reduction in this protein has been observed in patients with infertility and poor sperm motility [67,68,69,70].

The enriched classification was also employed to assess the potential functional roles of the DAPs. FASN and CLU were identified as participants in oxidoreductase activity and metabolic pathways, being more abundant in individuals from the wild. Both FASN and CLU have been recognized as potential sperm markers for recurrent pregnancy loss [70,71,72]. Lipids constitute a major component of teleost fish spermatozoa [73]. The fatty acid composition of spermatozoa is influenced by their diet [74,75]. The diet significantly alters the fertilizing capability of fresh sperm. The transfer of essential fatty acids from the diet to the semen is effective, and this transfer may exert biological effects on semen’s fertilizing capacity [76]. CLU is a small glycoprotein present in the seminal plasma and on the sperm surface of various species, including dogs [77], bulls [78], boars [79], rams [80], peccaries [81], camels [82], and coatis [46]. It is involved in sperm maturation, lipid transport, membrane remodeling, DNA repair, apoptosis inhibition, and cell cycle control [83,84,85,86]. CLU is secreted in response to cellular damage and heat shock and protects cells from these environmental stresses [46].

## 5. Conclusions

Differentially expressed proteins that play critical roles in spermatozoa could be key factors in explaining the limited reproductive success for species under managed care. This study provides new insights into characterizing the seminal plasma and spermatozoa proteomes in sharks. We observed a higher number of proteins in spermatozoa compared to seminal plasma, with minimal overlap between the two sample types. Moreover, we have shown that environmental conditions can lead to differences in protein composition in spermatozoa with minimal impact on seminal plasma proteins in *S. canicula.* While our understanding of the complete functions of the differentially abundant proteins is not yet complete, this study serves as a valuable foundation for further research. Specifically, these results will enable a deeper elucidation of the molecular mechanisms involved in this particular condition and may shed further light on the key sperm proteins implicated in fertilization, as well as for the development of effective reproductive technologies in conservation efforts for chondrichthyan species. 

## Figures and Tables

**Figure 1 animals-14-01281-f001:**
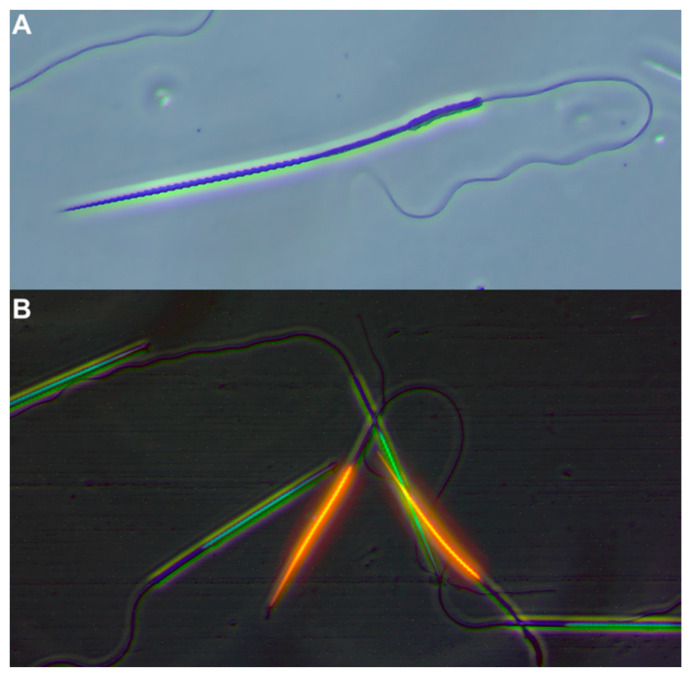
Representative images of small-spotted catshark (*Scyliorhinus canicula*) sperm cell quality. Scale bar, 20 μm. (**A**) Phase-contrast image of sperm cells at 400× magnification. (**B**) Phase-contrast and epifluorescence merged micrographs of sperm cells stained with SYBR-14 (green) and propidium iodide (red) at 200× magnification. Green fluorescence shows live sperm, and red fluorescence indicates dead sperm.

**Figure 2 animals-14-01281-f002:**
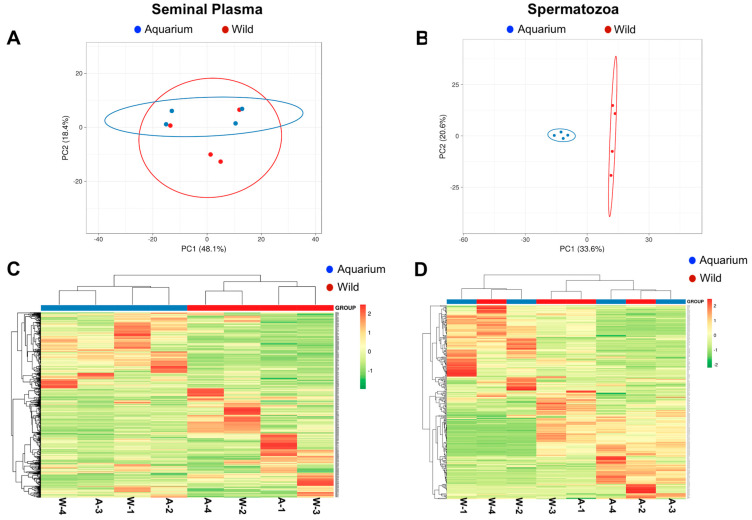
Principal component analysis (PCA) (left panels) showing the separation among animal origin for the seminal plasma (**A**) and spermatozoa (**B**) of total proteins. Blue color indicates wild animals. Red color indicates aquarium animals. Dots with similar color indicate technical replicates for each source (n = 4). Heatmap with dendrograms (right panels) representing the differentially abundant proteins among seminal plasma (**C**) and spermatozoa (**D**) from aquarium (A) and wild (W) animals. The data were obtained from 4 replicates for each source. The hierarchical clustering tree is shown at the top of the heat map. The relative expression level of the differentially abundant proteins is shown on a color scale from orange representing the highest level to green representing the lowest level.

**Figure 3 animals-14-01281-f003:**
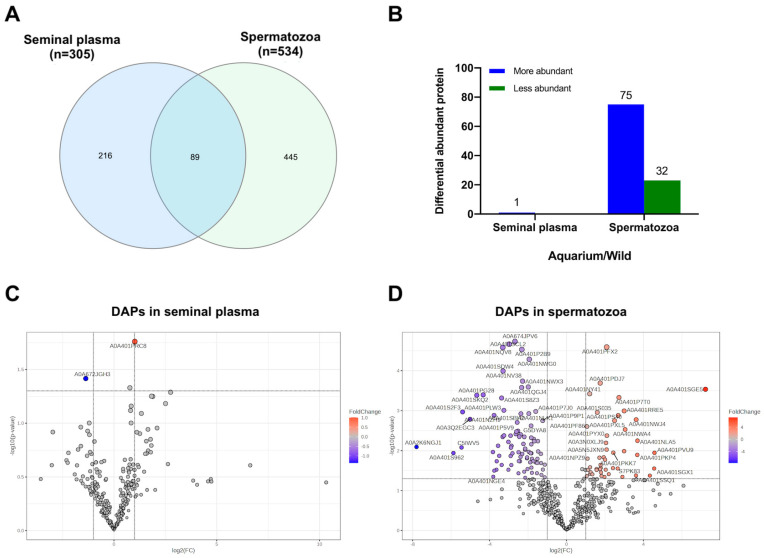
Differentially abundant proteins in seminal plasma and spermatozoa protein extracts from wild and aquarium samples of small-spotted catshark (*Scyliorhinus canicula*). (**A**) Venn diagram showing the number of unique proteins in seminal plasma and spermatozoa samples. (**B**) Number of proteins exhibiting significant differences within seminal plasma and spermatozoa between aquarium and wild groups. Visualizations of DAPs in volcano plots using Metaboanalyst are as follows: the x-axis represents logFC; the y-axis represents the −log10 of a *p*-value of <0.05. Proteins with logFC ≥ 1 are represented by red dots, and those with logFC ≤ −1 by blue dots. Significant DAPs are labeled with gene names. Black dots indicate proteins that did not show significant changes. (**C**) Volcano plot for seminal plasma proteins. (**D**) Volcano plot for spermatozoa proteins.

**Figure 4 animals-14-01281-f004:**
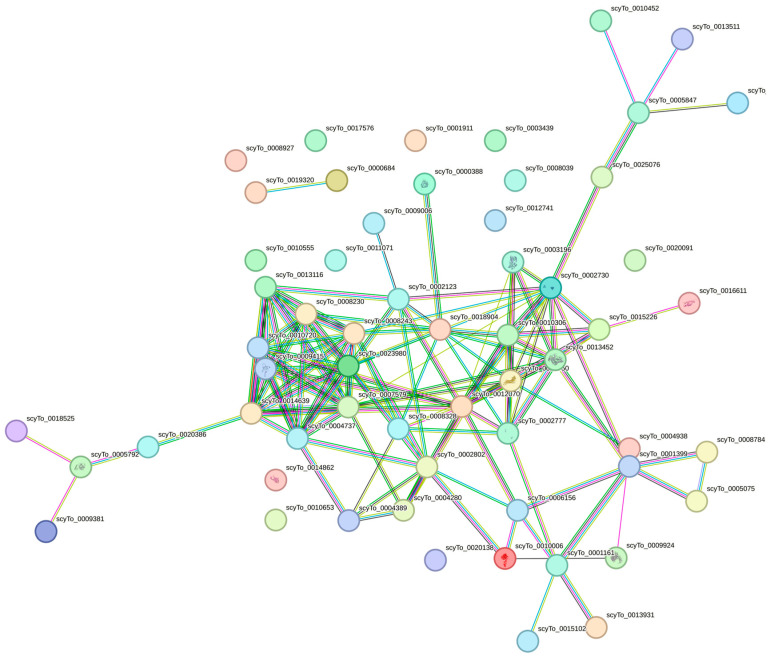
Protein-protein interaction networks of spermatozoa proteins differentially abundant between wild and aquarium semen annotated in the UNIPROT in cloudy catshark (*Scyliorhinus torazame*). One cluster of interacting proteins was identified using the STRING software (version 12. https://string-db.org, accessed on 1 March 2024) with a high confidence score. The line size indicates a high interaction score (tight lines indicate a high score > 0.7; thin lines indicate a medium score > 0.4). Each node represents a protein, whereas edges indicate the strength of the relationship between proteins (i.e., more edges indicate higher confidence). The proteins identified in small-spotted catshark spermatozoa of aquarium and wild males are shown in Table 2.

**Table 1 animals-14-01281-t001:** Bayesian analyses of the seminal parameters of small-spotted catsharks reveal differences between wild (W) and aquarium (A) animals, computed as W-A.

Traits	D_W-A_	HPD95	P0
Sperm concentration (10^6^/mL)	39.13	−20.8, 104.2	0.90
Motility (%)	−35.88	−78.2, 8.2	0.95
Viability (%)	−12.60	−46.5, 22.1	0.79
Mitochondrial membrane high potential (%)	−8.17	−20.5, 4.2	0.91

D_W-A_ = Mean of the difference W-A (median of the marginal posterior distribution of the difference between the W and the A groups). Relevant value (proposed as one-third of the SD of the trait), rounded to the first significant number. P0 = Probability of the difference (DW-A) being greater than 0 when DW-A > 0 or lower than 0 when DW-A < 0. HPD95% = The highest posterior density region at 95% probability. Statistical differences were assumed if |D_W-A_| surpassed the R value and its P0 > 0.80.

**Table 2 animals-14-01281-t002:** Differentially abundant proteins in the spermatozoa of wild (W) and aquarium (A) small-spotted catsharks (*Scyliorhinus canicula*), annotated in the UNIPROT database for cloudy catsharks (*Scyliorhinus torazame*).

No.	Protein Name	Accesion Number	Gene ID	FC A/W	*p*-Value
1	26S proteasome non-ATPase regulatory subunit 2	A0A401NY41	scyTo_0010006	1.2	0.002
2	Aconitate hydratase, mitochondrial	A0A401P1B5	scyTo_0012070	−1.5	0.04
3	Acylamino-acid-releasing enzyme	A0A401P232	scyTo_0000684	−2.2	0.022
4	ADP/ATP translocase (ADP, ATP carrier protein)	A0A401P5V9	scyTo_0015226	−2.6	0.011
5	Alcohol dehydrogenase-like N-terminal domain-containing protein	A0A401QCT0	scyTo_0023980	−3.0	0.021
6	Amidohydrolase-related domain-containing protein	A0A401NWA4	scyTo_0000388	3.0	0.014
7	ATP synthase subunit beta	A0A401PKK7	scyTo_0002730	2.4	0.033
8	Axonemal dynein light intermediate polypeptide 1	A0A401NUF3	scyTo_0011760	1.8	0.042
9	Band 7 domain-containing protein	A0A401NLA5	scyTo_0009381	3.7	0.023
10	Band 7 domain-containing protein	A0A401PXL5	scyTo_0018525	2.5	0.012
11	Band 7 domain-containing protein	A0A401PUL5	scyTo_0016611	1.1	0.035
12	Calponin-homology (CH) domain-containing protein	A0A401NWG0	scyTo_0014862	−1.9	0.000
13	Carboxylic ester hydrolase	A0A401NZH9	scyTo_0004938	−3.8	0.009
14	Clusterin	A0A401PF86	scyTo_0008927	1.1	0.003
15	CN hydrolase domain-containing protein	A0A401Q4C5	scyTo_0018904	2.0	0.023
16	CUB domain-containing protein	A0A401PXA0	scyTo_0019320	0.8	0.033
17	EF-hand domain-containing protein	A0A401PGM8	scyTo_0001911	−2.4	0.036
18	Endoplasmic reticulum resident protein 29	A0A401P7T0	scyTo_0013931	2.7	0.004
19	Enoyl reductase (ER) domain-containing protein	A0A401P5Y1	scyTo_0008243	−2.1	0.018
20	Enoyl-CoA hydratase	A0A401NRL5	scyTo_0014639	1.2	0.039
21	Fatty acid synthase	A0A401P5R6	scyTo_0008230	3.0	0.033
22	Fumarate hydratase, mitochondrial	A0A401PXE9	scyTo_0016750	−1.5	0.006
23	Glutathione transferase	A0A401PDJ7	scyTo_0008784	1.8	0.002
24	Glutathione transferase	A0A401P1T3	scyTo_0005075	−3.6	0.028
25	Glycerol kinase	A0A401PKW0	scyTo_0002802	−1.8	0.043
26	Glycerol-3-phosphate dehydrogenase [NAD(+)]	A0A401NP72	scyTo_0004280	1.8	0.048
27	Guanylin	A0A401P9P1	scyTo_0010653	−1.2	0.002
28	Heat shock protein 90	A0A401QGJ4	scyTo_0025076	−2.3	0.001
29	Hydroxyacyl-coenzyme A dehydrogenase, mitochondrial	A0A401NV38	scyTo_0007579	−2.3	0.002
30	IgGFc-binding protein N-terminal domain-containing protein	A0A401PYX0	scyTo_0020091	2.1	0.011
31	Importin N-terminal domain-containing protein	A0A401NW55	scyTo_0009924	−2.6	0.019
32	Inositol 1,4,5-trisphosphate receptor	A0A401PCL8	scyTo_0005792	1.7	0.029
33	Isopropylmalate dehydrogenase-like domain-containing protein	A0A401P3W7	scyTo_0010306	1.0	0.035
34	Malate dehydrogenase	A0A401NWX3	scyTo_0013452	−2.0	0.002
35	MARVEL domain-containing protein	A0A401P8H4	scyTo_0010555	−3.6	0.047
36	Medium-chain specific acyl-CoA dehydrogenase, mitochondrial	A0A401NPH1	scyTo_0013116	−1.3	0.048
37	Mesothelin-like protein	A0A401PMK1	scyTo_0003439	−2.0	0.035
38	Methyltransferase small domain-containing protein	A0A401PVU9	scyTo_0017576	4.6	0.036
39	Mucin-like protein	A0A401P6M2	scyTo_0010452	−1.3	0.014
40	NADH dehydrogenase [ubiquinone] flavoprotein 2, mitochondrial	A0A401PKP4	scyTo_0002777	3.7	0.039
41	Outer dynein arm-docking complex subunit 4 (Tetratricopeptide repeat protein 25)	A0A401PDC6	scyTo_0005847	−2.9	0.039
42	Phosphoglycerate kinase	A0A401PLW3	scyTo_0003196	−3.3	0.008
43	protein disulfide-isomeras	A0A401PA50	scyTo_0001161	1.9	0.025
44	Protein odr-4 homolog	A0A401P2P7	scyTo_0008039	−1.5	0.017
45	Renin receptor	A0A401NH33	scyTo_0011071	−1.3	0.024
46	Rieske domain-containing protein	A0A401PHR0	scyTo_0002123	−1.9	0.021
47	S-adenosyl-L-homocysteine hydrolase NAD binding domain-containing protein	A0A401PRC8	scyTo_0020386	−1.7	0.032
48	S-formylglutathione hydrolase	A0A401P7J0	scyTo_0008328	−1.6	0.001
49	Short chain dehydrogenase/reductase family 16C member 5	A0A401NWJ4	scyTo_0004737	3.6	0.011
50	SMP-LTD domain-containing protein	A0A401PFX2	scyTo_0009006	2.1	0.000
51	Sulfhydryl oxidase	A0A401P2B9	scyTo_0015102	−2.3	0.000
52	Superoxide dismutase	A0A401PG28	scyTo_0006156	−4.3	0.005
53	Synaptogyrin	A0A401NHV2	scyTo_0012741	−1.8	0.014
54	Thiolase N-terminal domain-containing protein	A0A401PAQ5	scyTo_0010720	−2.1	0.024
55	Thiolase N-terminal domain-containing protein	A0A401NLX1	scyTo_0009415	−2.0	0.003
56	Thioredoxin domain-containing protein	A0A401PCP3	scyTo_0001399	−3.8	0.035
57	Triokinase/FMN cyclase	A0A401NQV8	scyTo_0004389	−3.3	0.000
58	VWFA domain-containing protein	A0A401NYA7	scyTo_0013511	−2.3	0.026
59	ZP domain-containing protein	A0A401Q018	scyTo_0020138	−1.6	0.026

**Table 3 animals-14-01281-t003:** Gene ontology (GO) terms such as biological process, molecular functions, and KEGG pathways of differentially abundant proteins in small-spotted catshark (*Scyliorhinus canicula*) spermatozoa collected from wild and aquarium animals annotated in the UNIPROT in cloudy catshark (*Scyliorhinus torazame*).

Category	Term Name	Count	FDR
Biological process			
GO:0009987	Cellular process	53	0.0374
GO:0044281	Small molecule metabolic process	17	0.00020
GO:0019752	Carboxylic acid metabolic process	9	0.0474
GO:0006091	Generation of precursor metabolites and energy	8	0.00078
GO:0015980	Energy derivation by oxidation of organic compounds	7	0.00078
GO:0009060	Aerobic respiration	6	0.00078
GO:0006099	Tricarboxylic acid cycle	4	0.0026
GO:0006081	Cellular aldehyde metabolic process	3	0.0481
GO:0019563	Glycerol catabolic process	2	0.0475
Molecular Function			
GO:0003824	Catalytic activity	33	0.00038
GO:0016491	Oxidoreductase activity	16	8.74 × 10^−8^
GO:0042802	Identical protein binding	11	0.0486
GO:0016616	Oxidoreductase activity, acting on the CH-OH group of donors, NAD or NADP as acceptor	7	2.44 × 10^−5^
GO:0051287	NAD binding	5	0.00038
GO:0016836	Hydro-lyase activity	4	0.0065
GO:0003857	3-hydroxyacyl-CoA dehydrogenase activity	2	0.0150
Kegg Pathway			
map01100	Metabolic pathways	30	5.00 × 10^−10^
map01200	Carbon metabolism	10	3.80 × 10^−9^
map01212	Fatty acid metabolism	6	4.31 × 10^−6^
map00071	Fatty acid degradation	5	1.85 × 10^−5^
map00280	Valine, leucine and isoleucine degradation	5	3.62 × 10^−5^
map00310	Lysine degradation	6	3.62 × 10^−5^
map00650	Butanoate metabolism	4	3.62 × 10^−5^
map00020	Citrate cycle (TCA cycle)	4	0.00028
map00380	Tryptophan metabolism	4	0.00040
map00630	Glyoxylate and dicarboxylate metabolism	4	0.00040
map00620	Pyruvate metabolism	4	0.0012
map00640	Propanoate metabolism	3	0.0052
map00072	Synthesis and degradation of ketone bodies	2	0.0080
map01210	2-Oxocarboxylic acid metabolism	2	0.0198
map00980	Metabolism of xenobiotics by cytochrome P450	2	0.0393
map00982	Drug metabolism—cytochrome P450	2	0.0430
map01230	Biosynthesis of amino acids	3	0.0430
map00062	Fatty acid elongation	2	0.0457
map00900	Terpenoid backbone biosynthesis	2	0.0457

## Data Availability

The mass spectrometry proteomics data have been deposited in the ProteomeXchange Consortium via the PRIDE partner repository, with the dataset identifiers PXD046348 (SWATH data) and PXD046350 (Spectral Library data).

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
