# Peer review of "Proteomic Insights into Seminal Plasma and Spermatozoa Proteins of Small-Spotted Catsharks, Scyliorhinus canicula: Implications for Reproductive Conservation in Aquariums"

_animals, 2024, doi:10.3390/ani14091281_

Round 1
Reviewer 1 Report
Comments and Suggestions for Authors
Overall, the paper is solid and highly interesting, with significant implications for society and the conservation of wild animals. However, there are a few minor issues that need to be addressed to enhance the quality and ensure better clarity of the concepts.
1) In method section 2.5, 30ug of protein was digested with 600ng of trypsin (ratio of 50:1), while in differential expression analysis, it is 20ug protein with 800ng of trypsin, is there a reason change conditions?
2) In line 321, the author mentioned “Protein abundance was estimated using two Scaffold parameters: quantitative data and the number of unique peptides”. The author should provide further clarification regarding the methodology used to obtain/calculate the two parameters. These parameters are crucial as they serve as key criteria for all subsequent analyses.
3) In line 299-303, “1 and 42 respectively were not present in the Scyliorhinus torazame UNIPROT database” and only “59 DAPs” was further analyzed later. The “1 and 42” take around 40% of all DAPs, the author should have some discussion about those proteins too. Are they mis-analyzed or just not in database yet?
4) To validate the MS analysis result, it would be better if the author could demonstrate the increased protein abundance by western blot.
5) In the discussion part, when author analysis the DAP individually, it would be better to relate if they are decreased or increased abundance in aquarium, because the main topic (as mentioned in the title: reproductive conservation in aquarium) is to see the difference between aquarium or wild. Some examples: in line 428 and later, A0A401NWX3 and A0A401P3W7 are to key proteins, one is decreased and one is increased abundance based on Table 2. The author only discussed related pathways, it would be great if the author can also talk about why increase or decrease, and how des the quantity change affect the pathways and functions.
Author Response
Overall, the paper is solid and highly interesting, with significant implications for society and the conservation of wild animals. However, there are a few minor issues that need to be addressed to enhance the quality and ensure better clarity of the concepts.
Response
The authors would like to thank the referee for his/her careful reading of the manuscript and the comments. We have carefully edited this new version and corrected them.
Reviewer wrote:
- In method section 2.5, 30ug of protein was digested with 600ng of trypsin (ratio of 50:1), while in differential expression analysis, it is 20ug protein with 800ng of trypsin, is there a reason change conditions?
Response
Thanks for the comment. The use of 30 ug of protein digested with 600ng of trypsin was used for the spectral library building, while for the study of differential expression, 20ug protein with 800ng of trypsin were used, following the recommendations by the stablished protocol from the Proteomic Department at Universitat de Valencia, as described by Shevchenko A, Jensen ON, Podtelejnikov AV, Sagliocco F, Wilm M, Vorm O, Mortensen P, Boucherie H, Mann M.
Reviewer wrote:
- In line 321, the author mentioned “Protein abundance was estimated using two Scaffold parameters: quantitative data and the number of unique peptides”. The author should provide further clarification regarding the methodology used to obtain/calculate the two parameters. These parameters are crucial as they serve as key criteria for all subsequent analyses.
Response
This sentence should not be in the results section. It relates to the analysis of protein identification and validation, which involves both quantitative data and the number of unique peptides. This content has been removed from the Results section because it is intrinsically included in Section 2.6.
Reviewer wrote:
- In line 299-303, “1 and 42 respectively were not present in the Scyliorhinus torazame UNIPROT database” and only “59 DAPs” was further analyzed later. The “1 and 42” take around 40% of all DAPs, the author should have some discussion about those proteins too. Are they mis-analyzed or just not in database yet?
Response
This concerns the lack of information on these species. Specifically, 40% of all DAPs did not have any orthologs in other species. We have included this in a comment addressing the issue raised, as it indeed highlights the lack of information on these species. The sentence is as follows: “However, this study also highlights the lack of information on these species. Among the 2 DAPs identified in seminal plasma, only 1 has an ortholog (in either one-to-one or one-to-many relationships) in Scyliorhinus torazame. Conversely, of the 107 DAPs found in the spermatozoa, only 59 have orthologs in Scyliorhinus torazame. Specifically, for the spermatozoa, 42 DAPs did not have any orthologs in other species.”
Reviewer wrote:
- To validate the MS analysis result, it would be better if the author could demonstrate the increased protein abundance by western blot.
Response
This study is part of the initiation of a research line focusing on the use of reproductive tools to advance captive breeding in sharks. Unfortunately, the samples were entirely utilized in a previous study and in this one, leaving no biological material available for the suggested validation. The initiation of new studies is not immediate as it requires relevant permissions from the ethics committee to access the captive population. This makes it impossible to address the requested validation in the short term. Additionally, the availability and optimization of antibodies is another sensitive issue, as funding is required to develop the correct antibody, especially for this uncommon species. While, as the reviewer points out, validation would be important, there are numerous initial studies in proteomics on unconventional species that are published without this information.
Reviewer wrote:
- In the discussion part, when author analysis the DAP individually, it would be better to relate if they are decreased or increased abundance in aquarium, because the main topic (as mentioned in the title: reproductive conservation in aquarium) is to see the difference between aquarium or wild. Some examples: in line 428 and later, A0A401NWX3 and A0A401P3W7 are to key proteins, one is decreased and one is increased abundance based on Table 2. The author only discussed related pathways, it would be great if the author can also talk about why increase or decrease, and how des the quantity change affect the pathways and functions.
Response
Thank you for the comment. We believe that in this initial work, it is not feasible for us to explain how variations in the concentration of these proteins could influence changes in functions. Such an explanation would be overly speculative. We require a specific experimental design to understand how one or a few proteins may be impacting fertility. Indeed, at certain points in the discussion, we have included some comments, albeit very cautiously (line 480).
Reviewer 2 Report
Comments and Suggestions for Authors
The manuscript showed a interesting study. I made few comments about the text. One of the aspects I missed was the inclusion of images about the species and semen and spermatozoa. I understanding that the objective was not morphological in itself, but I believe that adding images makes the manuscript more dynamic.

Author Response
Reviewer wrote:
The manuscript showed a interesting study. I made few comments about the text. One of the aspects I missed was the inclusion of images about the species and semen and spermatozoa. I understanding that the objective was not morphological in itself, but I believe that adding images makes the manuscript more dynamic.
Response to Reviewer
The authors would like to thank the referee for his/her careful reading of the manuscript. Following the suggestions, we have added a figure (Figure 1) with two images that depict the appearance of the spermatozoa and their condition during the viability test. Additionally, the minor comments have been included in the manuscript.